# Inverse Game Theory for Stackelberg Games: the Blessing of Bounded Rationality

**Jibang Wu**
Department of Computer Science
University of Chicago
wujibang@uchicago.edu

**Weiran Shen**
Gaoling School of Artificial Intelligence
Renmin University of China
shenweiran@ruc.edu.cn

**Fei Fang**
Institute for Software Research
Carnegie Mellon University
feif@cmu.edu

**Haifeng Xu**
Department of Computer Science
University of Chicago
haifengxu@uchicago.edu

## Abstract

Optimizing strategic decisions (a.k.a. computing equilibrium) is key to the success of many non-cooperative multi-agent applications. However, in many real-world situations, we may face the exact opposite of this game-theoretic problem — instead of prescribing equilibrium of a given game, we may directly observe the agents' equilibrium behaviors but want to infer the underlying parameters of an unknown game. This research question, also known as *inverse game theory*, has been studied in multiple recent works in the context of Stackelberg games. Unfortunately, existing works exhibit quite negative results, showing statistical hardness [27, 37] and computational hardness [24, 25, 26], assuming follower's perfectly rational behaviors. Our work relaxes the perfect rationality agent assumption to the classic *quantal response* model, a more realistic behavior model of bounded rationality. Interestingly, we show that the smooth property brought by such bounded rationality model actually leads to provably more efficient learning of the follower utility parameters in general Stackelberg games. Systematic empirical experiments on synthesized games confirm our theoretical results and further suggest its robustness beyond the strict quantal response model.

## 1 Introduction

One primary objective of game theory is to predict the behaviors of agents through equilibrium concepts in a given game. In practice, however, we may observe some equilibrium behaviors of agents, but the game itself turns out to be unknown. For example, an online shopping platform can observe the shoppers' purchase decisions on different sale prices, but the platform has limited knowledge of the exact utilities of the shoppers. Similarly, while the policymaker could observe the market reactions to its policy announcement, the exact motives behind traders' reactions are usually unclear. In various security domains, the defender may want to understand the intentions or incentives of the attackers from their responses to different defense strategies so as to improve her future defense strategy. As such, recovering the underlying game parameters would not only lead us to better strategic decisions, but also improve our explications of the motives and rationale in the dark.

These potentials and prospects motivate a class of research problems known as the inverse game theory [26]: *given the agents' equilibrium behaviors, what are possible utilities that induce these behaviors?* In this paper, we specifically target the sequential game setting from the perspective of the

36th Conference on Neural Information Processing Systems (NeurIPS 2022).

first-moving agent (e.g., Internet platform, policymaker, or defender) whose different strategies (e.g., price, regulation, or defense scheme) would induce different equilibrium behaviors of the following agent (e.g., Internet users, traders, or attacker). Studies of such game settings have seen broad impacts and extensive applications ranging from the principal-agent problems in contract design [21, 19], the AI Economist [43] to security games modeled for social good [16]. We formalize our problem under the normal form Stackelberg game, where a leader has the commitment power of a randomized strategy, and a follower accordingly decides his response. It is known that the optimal commitment of the leader can be efficiently computed in a single linear program, given full knowledge of the game [14]. However, the inverse learning problem to determine the underlying game from the follower's responses is more challenging: Letchford et al. [27], Peng et al. [37] show that learning optimal leader strategy from the follower's best responses requires number of samples that is a high-degree polynomial in the game size and may be exponential in the worst cases. This significantly limits the practicality of these algorithms, as the leader usually cannot afford the time or cost to gather feedback from so many interactions.

More concerning is the inconvenient reality that we can hardly expect the agents' optimal equilibrium responses assumed in existing work. In fact, these shoppers, traders, or attackers themselves hardly know their exact utilities and are naturally unable to determine the expected-utility maximizing strategy. Extensive studies of behavioral economics and psychology [23, 5, 32, 11, 10] have pinpointed the cognitive limitations that make human decisions prone to the noisy perception of their utilities. Among various models for quantifying irrational agent behaviors, one of the most popular ones is perhaps the *quantal response* (QR) model [32], which adopts the well-known logit choice model to capture agents' probabilistic selection of actions. This will also be the bounded rationality model of our focus in this paper.

**The Blessing of Bounded Rationality.** The key insight revealed from this paper is that the extra layer of behavioral complexity due to bounded rationality, while complicating the modeling and computation, provides a more informative source for us to learn the underlying utility of agents. To understand the intuitions and motivations behind our results, consider a case where the follower has a dominated action $j_1$ as shown in Table 1, where the leader's and follower's utility of action profile $(i, j)$ is specified by $u_{i,j}, v_{i,j}$ respectively. Conventionally, such an instance is treated as a degenerated instance, because the leader could ignore the action $j_1$ that a perfectly rational follower would never play. Then, the optimal leader strategy is clearly to always play the action $i_2$. However, when facing a

| $u_{i,j}, v_{i,j}$ | $j_1$ | $j_2$ |
|---|---|---|
| $i_1$ | $100, 0.9$ | $0.9, 1$ |
| $i_2$ | $-99, 0.9$ | $1.1, 1$ |

Table 1: An example of dangerously "degenerated" Stackelberg game.

boundedly rational follower, it becomes possible to observe the response $j_1$ and estimate the utilities regarding this dominated action. For example, if the follower plays his action $j_1$ and $j_2$ at almost the same frequency, the follower's expected utility on the two actions should be close. Although such dominated action has no effect on the leader's optimal strategy against a perfectly rational follower, it could be a potentially damaging (or beneficial) action that leader want to avoid (or encourage) a bounded rational follower to play. That is, in the above game instance, if a somewhat irrational follower plays action $j_1$, it would be dangerous for the leader to play action $i_2$ yet rewarding to play action $i_1$; therefore, a more robust leader strategy should randomize by assigning some probability to play action $i_1$. We remark that in general, even without such extreme case of dominated actions, the extra payoff information is now available on how much worse (or better) it is to use the empirical frequency of the boundedly rational action responses (as long as some smoothness properties are exhibited), which are overlooked under the assumption of perfectly rational followers.

**Our Results.** We present a set of tight analysis on the number of strategies and sample complexity sufficient and necessary to learn the follower's utility, for both situations in which the leader can observe the follower's full mixed strategies or only the follower's sampled pure strategies. In the former situation of observing follower's mixed strategies, our algorithm can recover the follower utility parameters using $m$ follower mixed strategy responses in any general Stackelberg game where $m$ is the number of leader actions. Surprisingly, the required number of queries is independent of follower actions! This is due to the fact that the randomness introduced by bounded rationality carries much more information about follower payoffs, compared to the perfect best response. In the later

(more realistic) situation of only observing follower's sampled pure strategy, our algorithm learns the follower utility parameters within precision $\epsilon$ with probability at least $\delta$ using $\Theta\left(\frac{m \log(mn/\delta)}{\rho \epsilon^2}\right)$ carefully chosen queries, where $n$ is the number of follower actions and $\rho$ depends on agent's bounded rationality level and is of order $\Theta(1/n)$ for typical boundedly rational agents. Interestingly, the additional challenge of only observing sampled actions only deteriorates the sample complexity by a factor of $\log(mn)/\rho$.[1] These sample completexity results should be compared with that of [37, 27], which study similar learning questions but from perfectly rational follower responses. The $m \log(mn)/\rho$ order in our sample complexity is in sharp contrast to their complexity with *exponential* dependence in $m$ or $n$ in the worst case. Our experimental results empirically confirm the tightness of our sample complexity analysis.

At the conceptual level, our work illustrates that noises due to bounded rational behaviors could be leveraged as additional information sources to learn the follower utility. This intuition also drives the design of our analytical tools to explain how efficient and effective learning of the follower's utility is possible in real situations, in complementing the previous negative results developed under the idealized perfect rational behavior models [27, 37].

## 2 Related Work

**Learning in Stackelberg Game.** The learning problem in sequential games has been studied in several different setups. Marecki et al. [30], Balcan et al. [7] consider the online learning problem in the Stackelberg security game with adversarially chosen follower types. Bai et al. [6] consider a bandit learning setting where one could query any entry of the followers' utility under noise and use the estimation of utility to approximate the optimal leader strategy; however, this learning process assumes *centralization*, that is, the learner can control both leader's and follower's actions. More similar to ours is the strategic learning setup in Stackelberg games studied by [27, 37, 9], where the leader adaptively chooses her strategies based on the observation of the follower's best response and eventually recovers the follower's utility up to some precision level.

**Bounded Rationality.** McKelvey and Palfrey [32] introduced the quantal response equilibrium (QRE) by adopting the logit choice model [15, 31]. QRE serves as a strict generalization of Nash equilibrium (NE) — when the agents become perfectly rational, QRE converges to the NE. The modeling success of QR model attributes to the nice mathematical and statistical properties of the logit function that can capture a variety of boundedly rational behaviors under different parameter $\lambda$. QRE is widely adopted especially in Stackelberg (security) games [42, 35, 39, 16, 20, 12] and zero-sum games [29] and notably has been deployed in various real world application [2, 17]. Moreover, the model structure of QR has been also used in various other contexts, such as the softmax activation in neural network [18], multinomial logistic regression [8] and the multiplicative weight update algorithm for no-regret learning [3].

As an initial attempt to our general learning problem, we also adopt the QR model to capture our agent's bounded rational behavior, for its modeling success in practice and being the most common choice of prior work [42, 35, 16, 20, 12, 29, 2, 17]. We acknowledge that there exist other models of bounded rational behaviors beyond the QR model. For example, Kahneman [23] introduced the prospect theory to model the bounded rationality of agents in games under risk; Camerer et al. [11] proposed the cognitive hierarchy theory that classifies the agents according to their degree of reasoning in forming expectations of others. We anticipate that the message of our paper — i.e., the observation of suboptimal responses could provide additional information to learn the follower's preferences — would apply to many of these bounded rationality models.

**Inverse Game Theory.** Vorobeychik et al. [40] considered the payoff function learning problem using the strategy profiles and the corresponding utilities through regression. Kuleshov and Schrijvers [26] introduced the concept of inverse game theory, and the authors showed that the problem of computing the agents' utilities from a set of correlated equilibrium is NP-Hard, unless the game is known to have special structures. More recently, the inverse game theory problem is studied under the QR model and leads to a few positive results: Sinha et al. [39] considers the offline PAC-learning setup where the follower responses can be predicted with small error for a fixed leader strategy

---

[1]Note that the $\frac{\log(\delta)}{\epsilon^2}$ term comes from concentration bound and is natural when observations (i.e., observed follower actions) have randomness.

distribution; Haghtalab et al. [20] proves only three strategies are sufficient to recover linear follower payoff functions in security games; Ling et al. [29] presents an end-to-end learning framework that learns the zero-sum game payoff from its QRE. Following their success, our paper is the first work that provides theoretical guarantee of payoff recovery in *general* Stackelberg game. Finally, inverse problems have received significantly more attention in single-agent decision making problems; The most notable problem is the inverse reinforcement learning pioneered by Ng et al. [34], Abbeel and Ng [1].

## 3  Problem Formulation

**Game Setup**   We consider the Stackelberg game between a single leader (she) and follower (he). We let $U \in \mathbb{R}^{m \times n}$ (resp. $V \in \mathbb{R}^{m \times n}$) be the leader (resp. follower's) utility matrix, where $m, n$ are the number of actions for the leader (resp. follower). We use $\mathcal{G}(U, V)$ to denote the game instance. Each entry $u_{i,j}$ (resp. $v_{i,j}$) of the utility matrix denotes the leader's utility (resp. follower's utility) when leader plays action $i$ and follower plays action $j$. Without loss of generality, let $u_{i,j}, v_{i,j} \in [0, 1]$. Let $V_j \in \mathbb{R}^m$ be the $j$th column of the matrix $V$. We denote the set of the leader's (resp. follower's) action set by $[m] := \{1, \dots, m\}$ (resp. $[n] := \{1, \dots, n\}$).

In this sequential game, the leader moves first by committing to a (possibly randomized) strategy, $\boldsymbol{x} = (x_1, \cdots, x_m) \in \Delta_m$, where the simplex $\Delta_m = \{\boldsymbol{x} : \sum_{i \in [m]} x_i = 1 \text{ and } 0 \leq x_i \leq 1\}$ and each $x_i$ represents the probability the leader playing action $i$. Similarly, let $\Delta_n$ denote the follower's strategy space. Under perfect rationality, given the leader's committed strategy, the follower would in turns chooses the best response action $j^*$ that maximizes his utility, i.e., $j^* = \arg\max_{j \in [n]}\{\boldsymbol{x}^\top V_j\}$. In our problem, we use the QR model instead to capture the follower's bounded rational behavior. That is, the follower would respond to the leader's committed strategy by choosing an strategy $\boldsymbol{y}^*$ that maximizes his utility up to a Gibbs entropic regularizer, i.e., $\boldsymbol{y}^* = \arg\max_{\boldsymbol{y} \in \Delta_n}\{\lambda \boldsymbol{x}^\top V \boldsymbol{y} - \boldsymbol{y} \ln \boldsymbol{y}\}$. This is shown to be equivalent to the setting where the follower is best responding according to the payoff perturbed by noises from a Gumbel distribution [22]. And we know the close form solution of follower's optimal strategy for this convex optimization program is exactly the logit choice model on the true payoff, i.e., for each $j \in [n]$, $y_j^* = \frac{\exp(\lambda \boldsymbol{x}^\top V_j)}{\sum_{k \in [n]} \exp(\lambda \boldsymbol{x}^\top V_k)}$ [33].

We refer to $\lambda$ as the bounded rationality constant that is given in each specific problem, as several existing work have already determined its empirical value in practice: the human behavior experiments in [38, 41] compute $\lambda = 7.6$; the experiments [28, 36, 32] show $\lambda$ is in the range of 4 to 16.[2]

**Learning Problem**   We consider the inverse game theory problem in sequential game with unknown follower utility and seek to quantify how much the leader can learn about a bounded rational follower's utility. We frame this problem under an *active* and *strategic* learning setup, where the leader can interactively choose a randomized strategy and observe follower's strategic responses. Specifically, at each round $t \in [T]$, the leader commits to a strategy $\boldsymbol{x}(t)$. The follower observes the committed $\boldsymbol{x}(t)$ and responds based on the QR strategy $\boldsymbol{y}(t)$. Below we will consider both feedback settings based on whether the leader is able to observe the exact distribution $\boldsymbol{y}(t)$ or merely its samples.

We set our primary learning objective as to recover a full characterization of the follower's utility; our results below shall explain how it is unnecessary and almost unrealistic to expect an exact recovery of the follower's utility. And we show in Observation 1 and Theorem 1 that such utility characterization can be used to compute the optimal leader strategy under both perfect rationality, known as the strong Stackelberg equilibrium (SSE), and bounded rationality, known as the quantal Stackelberg equilibrium (QSE). And besides developing the optimal (or robust) leader strategies, we believe the recovered utilities are generally useful for our better understanding and reasoning of the followers' motives. However, given the limited scope of the paper, we focus on the inverse game theory problems and defer the problems regarding how to strategize using the knowledge of game (i.e., the typical game-theoretical problems) to related and future work.

Such learning problem has been considered in [20] specifically for Stackelberg security games, where the payoff is a strictly simplified single-dimensional linear utility function. Our paper overcomes the curse of dimensionality and answers the open question in recovering payoffs in the general Stackelberg game. On the other hand, Sinha et al. [39] showed a case of learning the nonparametric

---

[2]The $\lambda$ estimations are normalized to the utility scale in $[0, 1]$.

Lipschitz function (which includes the payoff function in the general Stackelberg game as a special case) in PAC-learning setup and they obtained a sample complexity exponential to the number of actions. Notably, the PAC-learning problem is fundamentally different from our active learning problem, as its learning guarantee is tied to the given data distribution and is not guaranteed to recover the follower's payoff.

# 4 Theoretical Results

## 4.1 Warm-up: Learning from Mixed Strategies

As a warm-up, we first consider a rather ideal case where the leader can directly observe the follower's mixed strategy $\boldsymbol{y}(t)$. In this case, it turns out that the leader would be able to perfectly recover the follower's payoff matrix from his responses to $m$ different strategies and thereby determine the her optimal strategy. Despite a seemingly intuitive result, its underlying rationale is actually not as straightforward. Specifically, many would raise the following doubt: the logit transformation is not bijective and thus its inverse mapping is not injective; in particular, it only gives us a system of at most $n-1$ different linear equations w.r.t. the follower's utility; one can check that if we add a constant to all entries of the utility matrix, the resulting probability distribution stays the same after the logit transformation. Thus, it should require more than $m$ such linear equation systems to recover a utility matrix with $m \times n$ unknown parameters. However, thanks to Observation 1, it happens that the follower's utility matrix can be fully characterized by $m \times (n-1)$ parameters that is essentially the difference of each column in the utility matrix. This somewhat coincidentally compensates the missing information on follower utility due to the logit transformation.

Knowing that $m$ strategies is the lower bound of this learning problem in general, below we will explicitly construct a learning algorithm that have the matching upper bound. To begin, a useful game-theoretic property of Stackelberg games is the following observation about the class of follower utilities that will induce the same leader and follower policies. While similar observation has been made in [20, 39], we also provide a formal proof in Appendix A for completeness.

**Observation 1** (Equilibrium Invariance under Payoff Transformation). *For any $\widetilde{V} \in \{V + \boldsymbol{c} \otimes 1_n | \boldsymbol{c} \in \mathbb{R}^m\}$, i.e., a row-wise shifted matrix of $V$, the follower's quantal response (resp. best response) policy to leader's committed strategy remains the same, and thus the optimal leader strategy in SSE or QSE remains the same.*

Observation 1 suggests that the row-wise shifted payoff matrix is just as good as the ground-truth payoff matrix in our setting. This essentially means that only the difference between action payoffs matters for the follower's policy. As such, we introduce a row-wise distance metric that accommodates such policy-invariant transformation to empirically measure the quality of the recovered follower utility.

**Definition 1** (Logit Distance). *We define a* logit distance *between the ground truth follower utility $V$ and the recovered follower utility $\widetilde{V} \in \mathbb{R}^{m \times n}$, $\Phi(V, \widetilde{V}) = \frac{1}{mn} \sum_{i \in [m]} \min_z \left\| V_i - \widetilde{V}_i - z \right\|_1$. Whenever the distance $\Phi(V, \widetilde{V}) = 0$, we say that the recovered follower utility is perfect.*

We next present a result that generalizes the well-known result, *three strategies to success in security games*, by Haghtalab et al. [20]. Notably, we identify a simple but fundamental condition (in terms of rank) necessary to recover the game payoffs, rather than the special distance conditions tailored to the structure of the security game as in [20]. The notion of rank has a clear physical meaning and we would later follow this theoretical insights to design learning algorithm to actively select leader strategies to query.

**Proposition 1** ($m$ Strategies to Success). *There exists a learning algorithm that can always perfectly recover the follower strategy from $m$ queries of the follower's mixed strategies.*

*Proof Sketch.* We pick $m$ linearly independent basis vectors for each $\boldsymbol{x}(t)$ in $m$ rounds and argue that the following optimization program can perfectly recover the follower's utility matrix $\widetilde{V}$.

$$
\begin{aligned}
\text{minimize} \quad & \sum_{t \in [m]} \left[ \log \sum_{j \in [n]} \exp z_j(t) - \boldsymbol{y}(t) \cdot \boldsymbol{z}(t) \right] \\
& \boldsymbol{z}(t) = \lambda \boldsymbol{x}(t)^\top \widetilde{V}, \quad\quad\quad\quad\quad \text{for } t \in [m].
\end{aligned}
\tag{4.1}
$$

We can see that the objective of the optimization program is a log-sum-exp function w.r.t. variables $\{\boldsymbol{z}(t)\}_{t\in[m]}$, which is convex. This means we can determine its minimizer set of $\{\boldsymbol{z}(t)\}_{t\in[m]}$. Meanwhile, the constraints of the optimization program gives a system of linear equation between $\{\boldsymbol{z}(t), \boldsymbol{x}(t)\}_{t\in[m]}$ and the variable $\widetilde{V}$. But the solution of $\widetilde{V}$ is not unique, as the minimizer set of $\{\boldsymbol{z}(t)\}_{t\in[m]}$ contains infinitely many elements. But it turns out that when $\{\boldsymbol{x}(t)\}_{t\in[m]}$ forms an linearly independent basis of $\mathbb{R}^m$, any solution $\widetilde{V}$ to the linear system given by any minimizer $\{\boldsymbol{z}(t)\}_{t\in[m]}$ are guaranteed to have $\Phi(V, \widetilde{V}) = 0$. We defer the full proof to Appendix B.

□

## 4.2 More Realistic Situations: Learning from Realized Actions

In this section, we consider the more challenging yet realistic scenario, where the leader is able to observe a single action from follower at each round, i.e., the best response w.r.t. his perceived utility under the Gumbel noise, or equivalently the realized action of the follower's quantal response strategy. It turns out that the intuitions from Section 4.1 still apply, and we are able to prove a strict generalization of these results. In particular, Theorem 1 strengthens Observation 1 in that learning the follower's utility up to some logit distance could also lead to an approximation of the optimal leader strategy under some mild condition given by Definition 2 in general Stackelberg games. Theorem 2 generalizes Proposition 1, as we showcase the sample complexity of our learning framework to recover the follower's utility in face of the follower's stochastic responses.

**Definition 2** (Inducibility Gap). *For any follower utility $V$, we define its inducibility gap as*

$$\sigma(V) := \min_{j \in [n]} \max_{\boldsymbol{x} \in \Delta_m} \min_{j' \neq j} \boldsymbol{x}^\top V[e_j - e_{j'}].$$

*That is, the maximum constant $\sigma(V)$ such that for any follower actions $j \in [n]$, there exists a leader strategy $\boldsymbol{x}^j$ that makes $j$ dominate any other action $j'$ by a margin of at least $\sigma(V)$, i.e., $\boldsymbol{x}^j V e_j \geq \boldsymbol{x}^j V e_{j'} + \sigma(V), \forall j' \neq j \in [n]$.*

If a game has small inducibility gap $\sigma$, then there must exsit two follower actions $j, j'$ such that the follower's utility for action $j$ can never be $\delta$ better than his utility for action $j'$, regardless of what strategies the leader play. In such cases, action $j$ is essentially dominated by $j'$ (up to at most $\delta$). It is not difficult to see that in such case with small $\delta$ it will be difficult to recover all the payoffs in such cases since action $j$ is expected to be played very rarely. This intuition is also reflected in our following two results.

**Theorem 1.** *Given a follower utility $\widetilde{V}$ with inducibility gap $\sigma(\widetilde{V}) > 5\epsilon$, we can construct an $O(\epsilon/\sigma(\widetilde{V}))$-optimal leader strategy for any game $\mathcal{G}(U, V)$ with logit distance $\Phi(\widetilde{V}, V) \leq \frac{\epsilon}{mn}$.*

*Proof Sketch.* We prove through an explicit construction. That is, given the estimate of the follower's utility $\widetilde{V}$, we construct a $\epsilon$-robust strategy $\boldsymbol{x} = (1 - \frac{3\epsilon}{\sigma(V)})\widetilde{\boldsymbol{x}}^* + \frac{3\epsilon}{\sigma(V)}\boldsymbol{x}^{\widetilde{j}^*}$ based on the SSE $(\widetilde{\boldsymbol{x}}^*, \widetilde{j}^*)$ of $\mathcal{G}(U, \widetilde{V})$ and the strategy $\boldsymbol{x}^{\widetilde{j}^*}$ such that $(\boldsymbol{x}^{\widetilde{j}^*})^\top V e_{\widetilde{j}^*} \geq (\boldsymbol{x}^{\widetilde{j}^*})^\top V e_{j'} + \sigma(V), \forall j' \neq \widetilde{j}^*$. We show this strategy is guaranteed to be an $(\frac{6\epsilon}{\sigma(\widetilde{V}) - 2\epsilon})$-SSE of the Stackelberg game $\mathcal{G}(U, V)$. The proof then relies on two key observations stated in Lemma 1.1 and 1.2: First, given that $\Phi(\widetilde{V}, V) \leq \frac{\epsilon}{mn}$ and $\sigma(V) > 3\epsilon$, the best response of a robust strategy $\boldsymbol{x}$ in game $\mathcal{G}(U, V)$ remains the same as that of a game $\mathcal{G}(U, \widetilde{V})$, and so is the leader utility. This means $\boldsymbol{x}$ gets at least $(1 - \frac{3\epsilon}{\sigma(V)})$ portion of SSE utility in $\mathcal{G}(U, \widetilde{V})$. Second, the difference between the SSE utility in $\mathcal{G}(U, V)$ and $\mathcal{G}(U, \widetilde{V})$ are bounded by $\frac{3\epsilon}{\sigma(V)}$. Meanwhile, even though $V$ is unknown to us, Lemma 1.3 shows that we can bound $\sigma(V) \geq \sigma(\widetilde{V}) - 2\epsilon$, so we can use $\sigma(\widetilde{V}) - 2\epsilon$ to substitute $\sigma(V)$. And this requires $\sigma(\widetilde{V}) \geq 5\epsilon$. □

Due to the space limit, we defer the full statement of the lemmas and proofs to the Appendix C. After restoring the connections between the logit distance and the leader's optimal equilibrium utility, we now show the relationship between the logit distance and sample complexity in the learning problem. We remark that by satisfying our full rank condition, this sample complexity result does not depend on any additional parameter on the distance of queried leader strategies, such as $\lambda, \nu$ in [20], both of which are only guaranteed to affect the sample complexity by polynomial (not necessarily linear) factors w.r.t. the number of targets.

**Theorem 2.** *It takes* $\Theta(\frac{m \log(mn/\delta)}{\rho \epsilon^2})$ *queries of the follower's quantal response to recover the follower's utility* $\widetilde{V}$ *within the logit distance* $\Phi(V, \widetilde{V}) = \frac{\epsilon}{\lambda}$ *with probability at least* $1 - \delta$, *where* $\rho$ *is the least non-zero measure among all of the follower's mixed strategies induced by leader's strategy queries during learning.*

This theorem is a strict generalization of Proposition 1 and we defer the full proof to Appendix D due to space limit. The high level intuition comes from the fact that $(1 - \epsilon)$-multiplicative approximation guarantee is translated to $\epsilon$ additive error after the logarithmic transformation using the approximation that for small positive $\epsilon$ close to zero, we have $\ln(\frac{1}{1-\epsilon}) = O(\epsilon)$. And to obtain such $(1 - \epsilon)$-multiplicative approximation of an mixed strategy, we use standard concentration results for a tight sample complexity bound. We formalize these statements and proofs in Lemma 2.1, 2.2.

**Lemma 2.1.** *There exists a learning algorithm that can recover the follower's utility* $\widetilde{V}$ *within the logit distance* $\Phi(V, \widetilde{V}) = O(\frac{\epsilon}{\lambda})$ *from* $m$ *queries of the* $(1 - \epsilon)$-*multiplicative approximation of the follower's mixed strategies.*

**Lemma 2.2.** *For any discrete distribution* $\boldsymbol{y}$ *with support size* $n$ *and the least non-zero measure* $\min_{i \in [n], y_i > 0}\{y_i\} \geq \rho$, *with* $\Theta(\frac{\log(n/\delta)}{\rho \epsilon^2})$ *samples, the corresponding empirical distribution* $\widehat{\boldsymbol{y}}$ *is an* $(1 - \epsilon)$-*multiplicative approximation to* $\boldsymbol{y}$, *with probability at least* $1 - \delta$.

### 4.3 A Learning Framework of Practicality

**PURE, Less is More**   The above results lead to a simple but provably effective method, PURE; the name comes from the fact that it only uses the $m$ different pure strategies in $\Delta_m$, $\{\boldsymbol{x}(t)\}_{t \in [m]}$. As specified in the proof of Theorem 2, it gathers the follower's sampled quantal responses of these pure strategies to estimate the corresponding empirical distributions $\{\widehat{\boldsymbol{y}}(t)\}_{t \in [m]}$ and solves for the $\widetilde{V}$ through the optimization program 4.1. While it is a seemingly naive learning algorithm, we would like to make a few crucial points on its unique advantages from both theoretical and practical perspectives.

Theoretically, we know PURE is guaranteed to perfectly recover the follower utility in the setting of Section 4.1. More importantly, when randomness is present, PURE guarantees that the estimation error measured by the logit distance is always bounded as $O(\frac{\epsilon}{\lambda})$; the Equation (D.1) in the proof of Theorem 2 suggests that the inverse of a general row-stochastic matrix $X$ and the error matrix $\beta$ could otherwise lead to possibly unbounded estimation error.

Meanwhile, we anticipate that the simplicity of PURE would be especially valuable to its applicability in practice. First, the randomized leader strategies in many applications are difficult to be implemented precisely, because the followers may not have the perfect estimation of the leader's distributions of randomization. This means that observing the follower's responses to randomized leader strategies could be more noisy in nature. Second, it might be inappropriate and possibly forbidden for the learner (e.g., an Internet platform or policy marker) to frequently change its strategies (e.g., prices or policies). Instead, the deployment of PURE only requires the learner to observe the responses of only a small number of pure strategies at the population level.

**PURE for Structured Games**   We remark that the learning framework of PURE could be tailored to the special structures in Stackelberg game. For example, let us consider a celebrated variant, known as the Stackelberg security game.[3] Namely, a leader (defender) commits to a randomized allocation of security resource to defend a set of $n(= m)$ targets from a follower (attacker). In turn, the follower observes this randomized allocation and picks a target to attack. Both the leader and the follower receive payoffs depending on the target that was attacked and the probability that it was defended. So in this case the follower utility can be expressed as linear functions, where each entry in vector $\boldsymbol{w}, \boldsymbol{b} \in \mathbb{R}^n$ denotes, respectively, the attacker's cost and reward on each target. When the leader defends each target with the randomized strategy $\boldsymbol{x} \in \Delta_n$, if the follower attacks the target $j$, he receives utility based on the cost w.r.t. the chance target $j$ is defended, and the reward for the attack, i.e., $V(\boldsymbol{x}, j) = w_j x_j + b_j$. Then, we can use the learning framework of PURE that only solves for the linear utility function parameters using the optimization program 4.2. This not only reduces the

---

[3]For simplicity, we here present a standard simplification of Stackelberg security game, where the resources allocation and scheduling constraints are ignored and the defender's strategy space is simply the simplex $\Delta_m$. Our method can be extended to security games under the general definition by carefully picking strategies on the vertices of the constrained strategy space.

number of parameters to be learnt but also directly gives the reward and cost parameters of each targets. Our empirical experiments below suggest a significantly faster error convergence rate once the structure insights is brought into the learning framework.

$$\text{minimize} \quad \sum_{t \in [d]} \left[ \log \sum_{j \in [n]} \exp z_j(t) - \widetilde{\boldsymbol{y}}(t) \cdot \boldsymbol{z}(t) \right]$$
$$\boldsymbol{z}_j(t) = \lambda(w_j x_j(t) + b_j), \qquad\qquad\qquad \text{for } j \in [n], t \in [T]. \tag{4.2}$$

**PURE-Exp for the Worst Cases** In certain situations, however, the followers could be more rational and the parameter $\lambda$ is larger than the standard estimation. Then, the follower's stochastic quantal response becomes rather deterministic, and the least non-zero measure $\rho$ decreases. Lemma 2.2 suggests that querying through simple pure strategies could become much less inefficient in obtaining the $(1 - \epsilon)$-multiplicative approximation of the actual strategy. Nevertheless, it turns out that we can introduce the "exploration and exploitation" principle here for the remedy, and we thus name such variant of PURE algorithm as PURE-Exp. Specifically, we introduce an exploration procedure to search for better strategies if an empirical estimation of the follower strategy tends to concentrate on a single action. We knew such strategy would contain more noise than information, as the error introduced by its multiplicative approximation ratio can be significant; reversing a one-hot distribution from logit transformation provides no information about the follower utility. In this case, we carefully replace it by a perturbed strategy from the original strategy. This ensures that the resulting strategy set after replacement still forms a full-rank matrix that ensures the invertibility necessary for a provably more effective recovery of $V$ in Theorem 2. Otherwise, the algorithm would continue to exploit the leader strategies to better estimate the follower responses. Our empirical experiments show substantial performance improvement by PURE-Exp in those extreme cases.

---

**Algorithm 1** PURE-Exp

1: **Input:** Game parameters $m, n, \lambda$, QR oracle $\mathcal{O} : \Delta_m \to [n]$ and optimization program $\mathcal{Q}$ based on the game structure.
2: **Initialization:** $\mathcal{X}$, a list of leader strategies where the $i$-th strategy $\boldsymbol{x}^{(i)} \leftarrow [e_i]_{i \in [m]}$; $\mathcal{Y}$, a list of empirical estimation of follower strategies w.r.t. $\boldsymbol{x}^{(i)}$; set $i \leftarrow 0$.
3: **for** $t = 0, 1, \ldots, T$ **do**
4:      Use leader strategy $\boldsymbol{x}^{(i)}$ from $\mathcal{X}$ to query for follower response $j \leftarrow \mathcal{O}(\boldsymbol{x}^{(i)})$.
5:      Update empirical estimation $\boldsymbol{y}^{(i)}$ of the follower's QR strategy to $\boldsymbol{x}^{(i)}$.
6:      **if** Probability mass of $\boldsymbol{y}^{(i)}$ concentrates on a single action **then**
7:          Sample a random perturbation $\widetilde{\boldsymbol{x}}$ from simplex $\Delta_m$.
8:          Replace $\boldsymbol{x}^{(i)}$ in list $\mathcal{X}$ by the new strategy $\boldsymbol{x}^{(i)} \leftarrow \frac{1}{2}\widetilde{\boldsymbol{x}} + \frac{1}{2}e_i$.
9:          Reset the empirical estimator $\boldsymbol{y}^{(i)}$ in $\mathcal{Y}$.
10:      **end if**
11:      Update $i \leftarrow (i + 1) \mod m$.
12: **end for**
13: Solve the optimization program $\mathcal{Q}$ for the best game parameters using $\mathcal{X}, \mathcal{Y}$.

---

## 5 Experiment

In this section, we seek to further understand the empirical implications of our learnability results. A major challenge when evaluating the learning performance is that the measures rely on the underlying ground truth utility. While there are several real world data collected in particular to understand the human behaviors and QR model [32, 38, 41, 35], they are sensitive, proprietary datasets in security domains that we are unfortunately unable to access. Moreover, these offline dataset only offer limited number of offline samples that can hardly be used in our active learning setup. Therefore, our experiments have to rely on synthesized game instances, from which we can construct oracles to respond to the active learning queries and accurately evaluate for the learning performance. As motivated in the previous section, we will use the logit distance in Definition 1 to empirically measure the quality of recovered follower utilities.[4] We start by investigating the empirical performance of PURE in games synthesized using several sets of different parameters below.

---

[4]Except the varying parameters, we control the parameters as $m = n = 10, \alpha = 0.2, \lambda = 8$ by default, and plot their average performance across 5 different randomly generated instances with the standard deviation illustrated in the error bars or the lightly shaded regions.

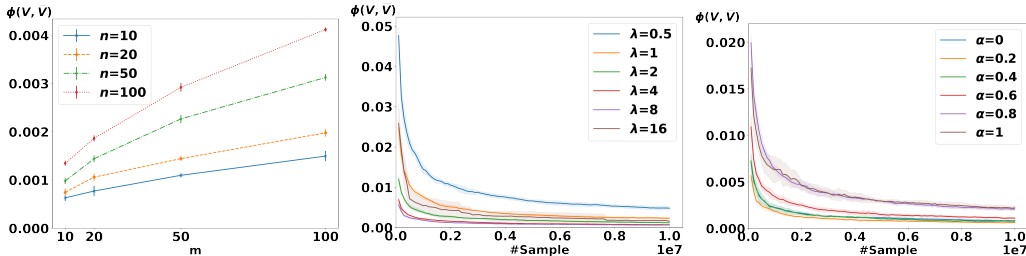

Figure 1: Recovering payoffs under varying parameters $m \times n$ (left), $\lambda$ (middle), $\alpha$ (right)

- **The number of leader and follower actions** $m, n$**:** We compare the learning performance in game of varying sizes, while fixing the number of query $T = 10^7$. In the left plot of Figure 1, the first trend to notice is that the error grows almost linear to $m$, exactly as Theorem 2 predicts. Meanwhile, the error also grows as $n$ increases, as the error bound depends on $1/\rho \geq n$. In Appendix, we shows that $1/\rho$ in average among those randomized generated game instances grows linearly with $n$, which justifies the almost linear relation between the logit distance and $n$.

- **The level of bounded rationality** $\lambda$**:** We consider different $\lambda$ ranging from 0.5 to 16 estimated in prior human behavior experiments [32, 38, 41]. In the middle plot of Figure 1, we display the convergence trend of logit distance in the number of queries. The PURE algorithm shows consistently good performance among these different $\lambda$. On one hand, in games with the smaller $\lambda$, the error tends to converge slower, as bounded by the $\frac{1}{\lambda \sqrt{t}}$ convergence rate implied by Theorem 2. On the other hand, the variance of error increases especially in the initial half of the timeline in games with larger $\lambda$. This is explained by the fact that sample complexity of learning distribution up to $(1 - \epsilon)$-multiplicative factor increases as the distribution concentrates when $\lambda$ increase.

- **The payoff margin** $\alpha$**:** We generate the follower's utility matrix, $V = \alpha I + (1 - \alpha)\Xi$, as a convex combination of diagonal matrix $I \in \mathbb{R}^{m \times n}$ and Gaussian random noise $\Xi$ normalized to $[0, 1]^{m \times n}$ such that the larger $\alpha$, the follower are likely to have higher margin for his best response against each of the leader's action. In the right plot of Figure 1, we can see a consistent trend of improving estimation of the follower's utility as query number increases across different level of $\alpha$. Interestingly, as the utility matrix becomes closer to the simple diagonal matrix, and the follower easily becomes less irrational, the convergence rate slows down; this again suggests our message on the *blessing of bounded rationality* that provides the stochasticity in follower's responses used as our additional information source.

We also compare the performance of PURE and its variants introduced in Section 4.3, and the results closely match with our theoretical insights. In the left plot of Figure 2, we compare PURE using only 10 leader strategies with the standard offline learning setup using $10^2, 10^3$ or $10^4$ leader strategies with less samples in average and less accurate estimation of follower response for each leader strategy. We can see that the PURE significantly outperforms these offline learning setups, especially when $\lambda$ is smaller such that the response of follower tends to be more irrational and thus "noisy". In the middle plot of Figure 2, we study the learning performance of PURE in various security games with or without using the optimization program specialized for the game structure (in dotted or straight lines). The result suggests that the structure insights can be used for fast recovery of follower utility. In the right plot of Figure 2, we found that PURE-Exp, with the principle of exploration and exploitation, are able to improve the learning performance in the case when the follower appears to be more rational. However, its performance also degrades as $\lambda$ further increases and the problem becomes almost the perfect rationality setting that are proved to be statistically hard to learn [27, 37]. In the limit of space, please check out Appendix E for more descriptions and analysis of our experiments.

## 6   Conclusion

Two common assumptions of a typical game theory problem are: (1) the agents know the game parameters; (2) the agents are perfectly rational. Though these assumptions have enabled elegant mathematical models and fundamental theoretical insights, they could be limiting in some real-world scenarios. Our paper tackles the particular problem in sequential game-theoretical interactions without these two common assumptions. While similar inverse game theory problems under perfect rationality

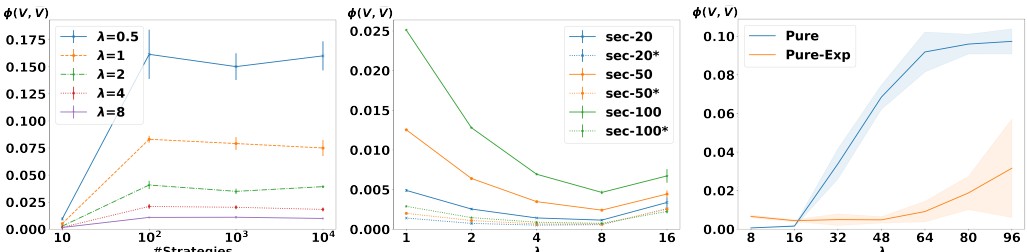

Figure 2: Comparision of `PURE` v.s. offline data (left), `PURE` with or without structure insights (middle), `PURE` v.s. `PURE-Exp` (right).

are shown to be statistically or computationally intractable, we made an intriguing finding in which relaxing us from these idealistic settings in turns lead us to a provably efficient learning guarantee. Therefore, we proposed the learning framework of `PURE` intended for fewer usage restrictions in real-world applications. In future work, we wish to extend our analysis and insights to more general game settings and other models of bounded rationality.

## Acknowledgement

We thank all the anonymous reviewers for their helpful comments. Co-author Haifeng Xu is supported in part by an ARO award W911NF-23-1-0030. Fei Fang was supported in part by NSF CAREER grant IIS-2046640. Weiran Shen gratefully acknowledges financial support from the National Natural Science Foundation of China (No. 62106273), the Fundamental Research Funds for the Central Universities, and the Research Funds of Renmin University of China.

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
