# OpenReview forum: "Inverse Game Theory for Stackelberg Games: the Blessing of Bounded Rationality"
_NeurIPS.cc/2022/Conference — NeurIPS 2022 Accept_

### Official Review · Reviewer_uvLt · 2022-07-04

**Rating:** 3
**Confidence:** 5
**Soundness:** 2 fair
**Presentation:** 2 fair
**Contribution:** 2 fair

**Summary:**

This paper studies learnability of approximate Stackelberg games in rational and quantal response follower setting. Results are mostly positive relying of ceratin properties of the utilities. The result explores sample complexities and performs experiments in simulation.

**Questions:**

I have none, but authors can respond to my comments above.

**Limitations:**

Not really applicable here.

**Strengths And Weaknesses:**

This content of this paper is incremental and to some extent covered in few related works, one of which[A] the authors do not include in references. The writing should provide more explanation and comparison to prior results.

First, the m strategies proof (Prop 2) is a minor extension of the work in the paper [20], same style of proof also except the utility function is slightly different.

Proposition 1 really does not deserve the importance given in the paper; it is a well-known fact for rational player games and for quantal response (stated in text in [A],[20]). That invariance can actually be easily handled by fixing one utility parameter to be 0, again as in [A], [20].

The paper [9, 37] gives an explicit best response sample complexity for computing approximate SSE in security games (with one resource, a security games is essentially a standard Stackelberg game with m=n). So, I do not understand how Thm 1 compares to this, I do not understand why Thm 1 assumes a given learned tilde(V) with a certain property? How can this be learned from best response samples? Thm 1 also refers to leader optimal strategy - should it be explicitly called SSE as written in the proof - this is important as the paper deals with both SSE and QSE.

In the same vein, [A] gives an explicit sample complexity for the quantal response case in achieving the same (m log m) term. How does this compare to Thm 2? In fact, that paper [A] present a more general result in terms of the capacity of the function class used in the exponent, making those result more applicable than the result here for bimatrix games.

Overall, I find the paper lacking sufficient novelty over prior work and also a lack of in-depth comparison with prior work.

Some citations are missing venue: e.g., [20].
Some citations seem irrelevant for Stackelberg games such as [24,25]


[A] Arunesh Sinha, Debarun Kar, Milind Tambe. Learning Adversary Behavior in Security Games: A PAC Model Perspective, in International Conference on Autonomous Agents and Multiagent Systems (AAMAS), May 2016

[9] Avrim Blum, Nika Haghtalab, and Ariel D Procaccia. Learning optimal commitment to overcome insecurity. Advances in Neural Information Processing Systems, 27, 2014.

[20] Nika Haghtalab, Fei Fang, Thanh Hong Nguyen, Arunesh Sinha, Ariel D Procaccia, and Milind Tambe. Three strategies to success: Learning adversary models in security games. 2016.

[37] Binghui Peng, Weiran Shen, Pingzhong Tang, and Song Zuo. Learning optimal strategies to commit to. In Proceedings of the AAAI Conference on Artificial Intelligence, volume 33, pages 2149–2156, 2019

---

> ### Author Response · Authors · 2022-08-02
> **Author Response**
>
> We thank the reviewer for the critical feedback and valuable suggestions. **However, we notice several important conceptual misunderstandings in the reviewer’s comments and thus respectfully ask the reviewer to re-evaluate the merit of this work.** Below are our clarifications and responses.
>
> First and foremost, we disagree with the reviewer's comments that our work is "incremental" and “lacking sufficient novelty”. The reasons are elaborated below:
>
> 1. Reference [A] studies a completely different learning setup from ours. Specifically, [A] is in the offline learning setup under PAC framework, so all their samples are i.i.d. drawn from a fixed distribution. Our learning setup is the active learning setup, in which the leader/learner can choose the queries (i.e., the mixed strategy), possibly based on previous feedback. We do appreciate the reviewer for pointing out this related reference (and will add a discussion on it), but we do not see how the PAC-learning results from [A] can "cover" (in fact even be comparable to) our results. And this is probably why the reviewer questioned how the leader can learn the follower's payoff matrices. In PAC learning, the leader is not guaranteed to learn the follower’s payoffs, since the small learning error in PAC only means the learned model is good under the given $(\mathcal{X},\mathcal{Y})$ distribution (e.g., an offline dataset). However, under active learning setup, the leader can indeed learn the payoffs due to the flexibility of choosing strategies in order to "probe" the uncertain directions. In fact, previous work [9,37] on active learning under perfectly rational follower responses can also learn follower's payoffs, up to some linear transformation.
>
> 2. [20] and [A] are limited to Stackelberg security games; their efficient learning algorithms heavily rely on the special structure of security games. More specifically, they exploited the fact that the attacker's utility at each action/target $i$ is a one-dimensional function mapping protection probability $x_i \in \mathbb{R}$  to a utility. In our problem, however, the follower's utility for each action is a high-dimensional function. Generalizing one-dimensional function learning to high-dimensional function learning is a well-known non-trivial learning task due to the curse of dimensionality. For example,  [A] does have a result for general follower utility function (their Thm. 3), but [A]'s sample complexity becomes EXPONENTIAL in the number of follower actions; in contrast, our sample complexity is almost linear. Therefore, we disagree with the comment that these previous results can "cover" our results.
>
> 3. Regarding comparison to [9,37], indeed they prove polynomial sample guarantees for learning optimal defender strategy from interacting with perfectly rational attackers. However, their best sample complexity (achieved in [37]) is $O(m^3)$, whereas instantiating our Thm 2 to the special case of security games already leads to an improved $O(m^2\log(m))$ complexity. Moreover, in general Stackelberg games, our $O(mn\log(mn))$ sample complexity is significantly better than the exponential sample complexity of [37]. Such accelerations are due to our learning from boundedly rational followers; this is precisely the key conceptual message our paper wants to convey, i.e., the "blessing of bounded rationality" as in our title.
>
> Regarding Prop 1 and Thm 1:
>
> We agree that Prop 1 is intuitive to experts of the game theory field. We did not mean to view it as a main result any way -- it is formalized as a proposition only for the sake of readability to the general audience since it is used later to introduce logit distance and thereby to characterize the solution set of recovered follower utilities used in Thm 1 and 2. In the revision, we will emphasize its connection to prior results and are happy to make it an *observation* if the reviewer prefers.
>
> However, our Thm 1 is presented to be a strengthened result over Prop 1, that is, how the learnt follower’s payoff (with potential errors) can be converted to an approximately optimal leader strategy. To our best knowledge, this result is novel and unknown in any prior work on learning Stackelberg equilibrium. In particular, this theorem introduces a new notion of "inducibility gap" that captures how difficult it is to convert the learned approximate payoff matrix to an approximately optimal leader strategy.
>
> Regarding Prop 2 and [20]:
>
> We do not view Prop 2 as our main result. This is why it is a *proposition* and why we intentionally draw a parallel to the established result in [20] (i.e., from 3 strategies to m strategies). We nevertheless stated it as a proposition since (1) it generalizes the main result of [20] to handle high-dimensional utility functions and thus has some merit (including generalizing their distance condition on query strategies); (2) more importantly, it prepares the reader with the intuitions for Thm 2, which is one of our main results.

---

> > ### Comment · Reviewer_uvLt · 2022-08-06
> > **Please update pdf and respond to my comments.**
> >
> > Sorry, I am little late in the discussion. First, I want to point out that NeurIPS allows updating the paper as a rebuttal revision to include new/modified things. I see that some other comments also asked for some discussion, etc. I do expect the authors to update the paper instead of saying that will add the discussion. For this purpose, to make space for discussion both the proof of Prop 1 (a standard known result) and Prop 2 can move to appendix.
> >
> > Next, for the comments for me. I agree that [A] is a PAC analysis in the batch mode, but [20] is not - in [20] the defender strategy being chosen depend on the other defender strategies. I agree with the authors' claim that compared to SSG with quantal reponse, the follower utility is higher dimension (and linear) in Stackelberg game. But again, I am left wondering what difference does it make in the proof/proof technique with the change from x_i (dependence on one component) to x (dependence on all components). The authors claim this is a generalization but what was hard about this generalization, or is a simple extension?
> >
> > Also, the claim that general utility leads to exponential sample complexity in [A] is informal. That exponential complexity is when the class of functions in the exponent is Lipschitz functions, whereas the class of functions here is linear. Please provide a more nuanced comparison - it is important to point out precisely how the active learning set-up helps.

---

### Official Review · Reviewer_osrx · 2022-07-11

**Rating:** 7
**Confidence:** 4
**Soundness:** 3 good
**Presentation:** 3 good
**Contribution:** 3 good

**Summary:**

Leader-follower games are among the most studied classes in the field of algorithmic game theory. The reason is simple: the corresponding solution concept – the strong Stackelberg equilibrium – proved to have many real-world applications. One of the central issues of Stackelberg equilibrium is its lack of robustness with respect to utilities. More specifically, it is imperative to correctly estimate the players’ priorities, represented through their utilities, otherwise the solution may force the leader to commit to a strategy that will eventually harm them.

In this paper, the authors present an approach that guarantees to uncover the follower’s true utilities in two-player leader-follower games through repeated interaction with the follower. The key appears to be the assumption of the follower’s bounded rationality. More precisely, the follower’s behavior needs to comply with the logit quantal response model of subrationality, playing each action with non-zero probability. Using this property, the authors are able to determine the number of interactions needed to estimate the follower’s utilities to a sufficient precision. Such characterization is possible not just when the leader observes the follower’s entire quantal-response strategy, but also in a more realistic setting, when they observe only samples from the strategy. Once the observations are collected, the follower’s utility is calculated as a solution of a particular mathematical program. The authors further show how to tailor their method to specific utility structures or when the follower’s quantal responses begin to approach best responses. The last part of the paper is dedicated to empirical analysis, depicting differences between true and estimated utilities over games with diverse sizes, players with different degrees of rationality, and utility matrices with varying levels of randomness.


**Questions:**

The authors of the original Quantal Response paper later introduced also other possible quantal-response models besides the logit one, e.g., probit, Luce, etc. Would this approach generalize to those models as well?

The definition of SSGs in section 4.3 appears to be a bit simplified for the sake of arriving at linear utilities. If I am not wrong, it is much more common to define defender strategies in SSGs using limited resources allocated to a subset of possible targets, expressing utilities using marginal coverage probabilities of the targets. Would this definition of SSGs lead to a similar result?

I liked how explicitly the experimental section compares PURE and PURE-EXP algorithms. Still, I am missing some intuition on when exactly it becomes beneficial to resort to PURE-EXP. Could the authors construct some explicit example when it is to? For example, on some fixed 2x2 game, what would be the follower’s rationality parameter (and the corresponding expected utilities of actions and quantal-response strategies) when the quantal responses approach best responses too closely to use PURE efficiently?

Could the authors comment on how do the computational times scale with the game size, i.e., how large games and datasets are solvable within some reasonable time limit?

Could the authors mention some possible applications of this exact learning method they propose? In which settings could the leader afford to practically iterate over their own strategies, when some could arguably lead to terrible outcomes?

Do the authors foresee how difficult it would be to generalize their approach into sequential games? More specifically, do they see some way how to deal with the problems associated with sequential reasoning, e.g., the existence of non-credible threats, in their learning framework?

Could the authors state if or when is the mathematical program from Theorem 2 solvable/approximable in polynomial time?


**Limitations:**

The authors did not include an explicit section about limitations, but they do mention some shortcomings in the introduction. Still, there are some limitations which most readers recognize when reading the text the authors may consider discussing. For example, how realistic is it to assume the follower’s model is fully known and what to do when it is not (perhaps something akin to playing a game with known utilities?)?


**Strengths And Weaknesses:**

I enjoyed reading the paper. Its motivation is clear and the authors describe well how their work fits into the literature on the topic. The results appear to be original and novel. The notation is not excessively complex and all notions are sufficiently well described. The arguments presented throughout the paper seem reasonable, facilitating the understanding of the exposition. I went through some of the proofs in the appendix and even though I did not go into all the details, as far as I can tell, they give an impression to be correct to me. Moreover, the results the authors present in the experimental section seem to corroborate the authors’ theoretical claims, even though at times the error bars happen to be rather larger (e.g., in the right graph of Fig. 3, for pure-exp and larger lambda), which would suggest averaging over more seeds might be a good idea.

Otherwise, I found it a bit confusing to refer to this setting as to a “sequential game”, which is a term commonly reserved for non-trivial extensive-form games, rather than a repeated Stackelberg (or leader-follower) normal-form game, which seems to be the case studied by the authors here.

For some of this work’s (possible) weakness, please see the next section.

Overall, I believe this is a nice work that may profit from some more detailed discussion regarding its limitations, but is worth publishing.

Nits:
Line 10: this work relax -> relaxes
Line 243: the equation in Def2 is missing a period at the end of the sentence
Line 259: we defer to the full statement -> we defer the full statement

---

> ### Author Response · Authors · 2022-08-02
> **Author Response**
>
> We thank the reviewer for the encouraging feedback and valuable suggestions! We will make sure to fix the typos in the next version. Below we respond to the technical questions:
>
> - Regarding generalizability to variants of quantal response models.
>
> We expect our framework to generalize to other bounded rationality models. Moreover, we believe the key insight of this paper will still hold, i.e., boundedly rational behaviors can help to ease the estimation of agents' utility parameters due to the "smoothed" agent response function. However, the rigorous theoretical analysis is likely to depend on the specific model at hand and, in some cases, may require different parameter estimation approaches other than the maximum likelihood. Our algorithm hinges on the nice mathematical structure of this original and popular quantal response model.
>
> - Regarding the definition of SSGs in section 4.3:
>
> Yes, our SSG definition in section 4.3 is slightly simplified, and we are fully aware that there is a more general framework for SSG with multiple resources and scheduling constraints. Exactly as the reviewer commented, our choice of the simpler definition is to make our structured learning approach easier to understand, since section 4.3 is just to demonstrate the flexibility of our framework for accommodating these structure constraints. Notably, such simplification is used in several other recent works as well [7,9,20,29,37]. And yes, we expect our framework to accommodate the general SSG model as well, given the resources and scheduling constraints can be formulated into additional linear constraints.
>
> - Regarding the usage of PURE-EXP:
>
> Yes, PURE-EXP becomes a better approach, when the agents tend to be strictly rational (i.e., best responding), and empirically when the $\lambda$ gets unusually large (e.g., >100). This is of course an extreme case, given $\lambda$ is measured to be around 8 in practice [28, 32, 36, 38, 40].
>
> - Regarding the computational time complexity:
>
> The sampling procedure takes $O(T)$ time. And once the distribution estimation is obtained, the complexity to compute optimized parameters is a polynomial function of the size of the utility matrix $(m,n)$.
>
> - Regarding the generalization to sequential games:
>
> Generalizing to sequential games could be challenging, as it might require learning arbitrary utility functions and the agents could have long-term objectives. The only work in this domain that we are aware of is, *Learning in Stackelberg Games with Non-myopic Agents, By Nika Haghtalab, Thodoris Lykouris, Sloan Nietert, Alex Wei. EC 2022*. So they are indeed exciting future directions for our study.
>
> - Regarding the time complexity of solving the optimization program in Theorem 2:
>
> The mathematical program in Theorem 2 is a well-known optimization problem with a convex objective and linear constraints. So not only theoretically there are polynomial time algorithms to approximate the optimal solution, but also in practice there are matured solvers to compute the solution efficiently. And there is our implementation in the supplementary materials.

---

> > ### Comment · Area_Chair_4MQr · 2022-08-04
> > **regarding simplified SSGs**
> >
> > > Exactly as the reviewer commented, our choice of the simpler definition is to make our structured learning approach easier to understand, since section 4.3 is just to demonstrate the flexibility of our framework for accommodating these structure constraints.
> >
> > It's perhaps a bit misleading to claim in the paper that you are applying it to SSG when you are actually applying it to a simplified version; for one your formulation removes the usual difficulty of SSGs which is dealing with the combinatorial space of possible allocations. I think this should at the very least be discussed. If you want to claim that this is done in the name of presentation then it would probably be best to show in the appendix that the more general version also works. Otherwise I would suggest that a phrasing more along the lines of "while we study a simplified SSG model here, we expect our framework to apply to standard SSGs as well, for XYZ reasons."
> > This is made worse by the fact that the paragraph itself talks about "security resources" in plural, but then implicitly simplifies to a single resource by stating $x\in \Delta^n$, with no comment that this simplification was made.

---

> > > ### Author Response · Authors · 2022-08-05
> > > **Author Response on Simplified SSGs**
> > >
> > > Thanks for the followup comment here. We fully agree with this suggestion and will make sure to clarify on these nuances. We note that, though the paper uses a simplified definition of SSGs,  the optimization program (4.2) can be extended for learning in general SSGs, since it only relies on the linearity of the payoff structure $V(x,j) = w_j x_j + b_j$. However, the caveat here is that we shall carefully query defender strategies that satisfy the additional resource allocation constraints of general SSGs. This turns out to only be a small adjustment to our active learning algorithm, e.g., to pick strategies on the vertices of constrained simplex space. We will include additional descriptions and experiments on this setting in the appendix.

---

### Official Review · Reviewer_AWq9 · 2022-07-11

**Rating:** 8
**Confidence:** 4
**Soundness:** 3 good
**Presentation:** 3 good
**Contribution:** 3 good

**Summary:**

This paper analyzes the problem of identifying follower utility in Stackelberg games, under the assumption that the follower *quantally* responds via softmax rather than playing a strict best response.  Remarkably, this turns out to be an easier problem.  The paper analyzes the sample and query complexity of identifying follower utility first under the toy assumption that the follower's full mixed strategy is observable, and then under the more realistic assumption that only actions realized from this strategy will be observed.  A specific algorithm is specified and used to validate the theoretical results empirically in experiments.

**Questions:**

1. p.8 L355: "We generate the follower's utility matrix $V=\alpha I + (1-\alpha)\Xi$": This seems like a very special kind of structure for the utility; should I be worried that the empirical results depend upon it?

**Limitations:**

I don't forsee any serious negative impact from this work.

**Strengths And Weaknesses:**

The Stackelberg setup is commonly used in security work, including with the assumption of quantal follower response, but it typically makes the assumption that follower utilities are known.  This paper studies the extension of this well-studied setting to a more realistic setting in which follower utilities must be inferred from their behavior.  These results are both novel, and likely to have significant impact on the security games literature.

The paper is well-written; the motivation, results, and proofs are all very clearly stated.  There are a few notational inconsistencies that I have noted below, but overall the paper was a pleasure to read, and did an excellent job of expressing some subtle distinctions.


__Minor issues__
- p.4: "And we know the close form of follower's optimal strategy... [33]": [Luce 1959] "Individual Choice Behavior" or [Train 2009] "Discrete Choice Methods with Simulation" would be more standard references for this equivalence.
- eqn (4.1): Why does the optimization use $\hat{y}$ instead of $y$?  This section assumes that we are able to observe the full mixed strategy, no?
- p.6 L251: "the strategy $x^{j^*}$": Shouldn't this be $x^{\tilde{j^*}}$?
- also L251: "for all $j' \ne j$": Should this be $j' \ne j^*$?
- p.7 L281: The empirical distributions are introduced as $\tilde{y}(t)$, but the program in equation (4.2) seems to use $\hat{y}(t)$, are these the same thing?
- The empirical distributions $\tilde{y}(t)$ (and/or perhaps $\hat{y}(t)$) in this section will all be deterministic/one-hot distributions, right?  Might not hurt to say so explicitly.
- Alg.1 line 7: "Sample a random perturbation $\tilde{x}$  from simplex $\Delta_m$": Does the sampling distribution matter here?  Should I just imagine a uniform sample?
- Alg.1: The description (lines 2--5) of how the $\mathcal{X}$ and $\mathcal{Y}$ lists are generated is kind of confusing; the initialization section makes it seem like they are generated up front, but it looks like $\mathcal{Y}$ at least is generated "on the fly" on line 5.

---

> ### Author Response · Authors · 2022-08-02
> **Author Response**
>
> We thank the reviewer for the encouraging feedback and valuable suggestions! We will make sure to improve our citation and notations in the next version. Below we respond to the few technical questions:
>
> - Regarding Alg.1 line 7:
>
> Yes, it is an uniformly random perturbation, just like the popular $\epsilon$-greedy, DBGD algorithms.
>
> - Regarding Alg.1:
>
> The $\mathcal{X}$ is initialized to contain a list of simplex vertices. The algorithm is designed to search and replace the existing leader strategies in the list with which the induced follower response distributions are easier to estimate.
>
> - p.8 L355 regarding utility generation in experiments:
>
> This way of generating utility matrices is intended to capture fully random payoffs (\alpha=0), fully structured payoffs (\alpha=1), as well as those in-between. Besides random payoffs, we are not aware of any other systematic way to generate representative payoffs for general Stackelberg games. Our experiments were actually designed to go beyond purely random cases and fully structured cases and thus used the above interpolation. However, if the reviewer may have any other suggested utility generation approach, we are more than happy to test on that as well!

---

### Meta-Review · Area_Chair_4MQr · 2022-08-20

**Recommendation:** Accept
**Confidence:** Certain

**Metareview:**

High-level view: this paper presents some interesting observations around
learning against a Stackelberg follower that corresponds to a quantal response model.
The learning seemingly relies strongly on the follower being a quantal responder with a logit regularizer, but this is an interesting setting to study, and one that seems to have been overlooked in the literature.
Thus I am broadly in favor of the paper.


Now, on a more detailed level, I do disagree with some claims made in the paper regarding bounded rationality. I would expect the authors to handle bounded rationality versus logit QRE more carefully in the camera ready. Personally, I would say that even the title of the paper ought to be changed. The short of it is that the authors conflate bounded rationality and quantal response behavior, but these are not the same thing. In fact, both the AC and other reviewers feel that the results are *not* likely to extend to most other models of bounded rationality. They may extend to other models of quantal response behavior, though.

Now, in more detail:
The authors state

> We remark that in general, even without such extreme case of dominated actions, the extra payoff information are now available on how much worse (or better) using the empirical frequency of actions from the boundedly rational responses, which has been overlooked due to the assumption of perfectly rational follower

But this can't be true: a special case of bounded rationality is that the
boundedly-rational player is rational enough to rule out the dominant action in
the example; in that case bounded rationality does not let us learn anything
from that action! The reason the authors can learn here is because they study a
very specific type of bounded rationality: quantal follower response.
Learnability then becomes doable specifically because the follower plays all
actions with non-zero probability.


A second thing is that the authors call it "perhaps surprising" etc that bounded
rationality helps. However, again, the authors are looking specifically at
quantal response behavior under the logit response model, which is well-known to be easier to handle. In
particular, adding the logit quantal response assumption corresponds to adding a
strongly-convex regularizer to the follower's best response problem. Regularizing
a nonsmooth function is known to lead to much nicer behavior in many settings,
e.g. in optimization, learning, etc. Moreover, the logit model specifically leads to a form of regularization that plays every action with non-zero probability, again a very useful fact for learning the follower model.

Finally, the authors say

> While our quantitative results do rely on the form of this model, we believe
53 the insight revealed from our model generalizes to most problems of learning from boundedly rational
54 agent behaviors.

This seems very unlikely, given what I stated above. Quantal response is a very
special type of bounded rationality. The results should probably only be
expected to extend to other problems that share some of these characteristics,
e.g. playing every action with non-zero probability, and possibly also corresponding to strongly
convex regularization.

Let me finish by reiterating that this is not a huge issue for the paper: the logit QRE setting is definitely important and of interest in its own right. But it's important to better delineate where we may hope for these results to generalize or not generalize.

**Award:**

No

---

### Decision · Program_Chairs · 2022-09-14

Accept